# Persistence of Chronic Lymphocytic Leukemia Stem-like Populations under Simultaneous In Vitro Treatment with Curcumin, Fludarabine, and Ibrutinib: Implications for Therapy Resistance

**DOI:** 10.3390/ijms25041994

**Published:** 2024-02-07

**Authors:** Àngel Bistué-Rovira, Laura G. Rico, Jorge Bardina, Jordi Juncà, Isabel Granada, Jolene A. Bradford, Michael D. Ward, Roser Salvia, Francesc Solé, Jordi Petriz

**Affiliations:** 1Departament de Farmacologia, Terapèutica i Toxicologia, Universitat Autònoma de Barcelona (UAB), 08193 Cerdanyola del Vallès, Spain; angel.bistue@uab.cat; 2Germans Trias i Pujol Research Institute (IGTP), Universitat Autònoma de Barcelona (UAB), 08916 Badalona, Spain; laura.garcia@bd.com (L.G.R.); rsalvia@igtp.cat (R.S.); 3Vall d’Hebron Institute of Oncology (VHIO), 08035 Barcelona, Spain; 4MDS Group, Institut de Recerca Contra la Leucèmia Josep Carreras, 08916 Badalona, Spain; jjunca@iconcologia.net (J.J.); igranada@iconcologia.net (I.G.); fsole@carrerasresearch.org (F.S.); 5Thermo Fisher Scientific, Fort Collins, CO 80524, USA; cytometryjo@gmail.com (J.A.B.); mike.ward@thermofisher.com (M.D.W.)

**Keywords:** curcumin, CLL, LSCs, flow cytometry, stem-like, alkaline phosphatase

## Abstract

Leukemic stem cells (LSCs) possess similar characteristics to normal hematopoietic stem cells, including self-renewal capacity, quiescence, ability to initiate leukemia, and drug resistance. These cells play a significant role in leukemia relapse, persisting even after apparent remission. LSCs were first described in 1994 by Lapidot et al. Although they have been extensively studied in acute leukemia, more LSC research is still needed in chronic lymphocytic leukemia (CLL) to understand if reduced apoptosis in mature cells should still be considered as the major cause of this disease. Here, we provide new evidence suggesting the existence of stem-like cell populations in CLL, which may help to understand the disease as well as to develop effective treatments. In this study, we identified a potential leukemic stem cell subpopulation using the tetraploid CLL cell line I83. This subpopulation is characterized by diploid cells that were capable of generating the I83 tetraploid population. Furthermore, we adapted a novel flow cytometry analysis protocol to detect CLL subpopulations with stem cell properties in peripheral blood samples and primary cultures from CLL patients. These cells were identified by their co-expression of CD19 and CD5, characteristic markers of CLL cells. As previously described, increased alkaline phosphatase (ALP) activity is indicative of stemness and pluripotency. Moreover, we used this method to investigate the potential synergistic effect of curcumin in combination with fludarabine and ibrutinib to deplete this subpopulation. Our results confirmed the effectiveness of this ALP-based analysis protocol in detecting and monitoring leukemic stem-like cells in CLL. This analysis also identified limitations in eradicating these populations using in vitro testing. Furthermore, our findings demonstrated that curcumin significantly enhanced the effects of fludarabine and ibrutinib on the leukemic fraction, exhibiting synergistic effects (combination drug index, CDI 0.97 and 0.37, respectively). Our results lend support to the existence of potential stem-like populations in CLL cell lines, and to the idea that curcumin could serve as an effective adjuvant in therapies aimed at eliminating these populations and improving treatment efficacy.

## 1. Introduction

Leukemic stem cells (LSCs) are leukemia-initiating cells that share characteristics with normal stem cells, including self-renewal capacity, quiescence, and resistance to chemotherapy, radiotherapy, and other external stimuli that induce apoptosis or cell differentiation. Despite their low proliferation rates when compared with committed progenitors, they are characterized as having high autorenewal capacity and ability to replenish the bulk of leukemia cells [1,2]. Leukemia stem cells are thought to play a major role in relapse, even after apparent complete remission has been achieved. A small number of these cells can remain undetected and regenerate the disease, causing relapse after treatment [3]. These populations have been described in some leukemias, such as chronic myeloid leukemia and acute myeloid leukemia [3,4], while their existence remains unclear and is not fully studied in others like chronic lymphocytic leukemia (CLL) [5,6].

In general, the characterization of stem cells presents some difficulties, such as their determination by immunophenotypic markers, as may be the case of CD34 for hematopoietic cells [3,7], with limited efficacy based on rarity. Given their own plasticity, there are a limited number of markers for these cells [1,3], and many of those used to isolate them are not multipotency indicators [3]. In addition, many of these markers can exhibit important variations depending on different factors, such as the activation state or characteristics of the cellular environment [3,8]. Some studies indicate that stem cells may be a heterogeneous population defined by function rather than a specific, uniform cellular entity characterized by specific markers [2,8]. For this reason, it is essential to have alternatives to identify these populations. One of these alternatives is functional analysis, which uses cell physiology instead of phenotypic markers to determine cell populations [9,10,11,12].

CLL is a hematological disease of high incidence in Western countries, caused by the abnormal expansion of B cells, which can produce pathological infiltration in organs and immunosuppression [13,14,15]. CLL is usually considered to arise due to reduced levels of apoptosis in mature B lymphocytes, leading to the accumulation of new mutations [16,17]. Consequently, residual levels of leukemic cells remain after treatment, which are usually characterized as being resistant to treatment and are associated with leukemia relapse. Measurable residual disease (MRD) is the presence of leukemia cells below the level that can be detected by routine tests [18]. MRD detection by using flow cytometry is currently a test with high prognostic value, as patients with negative MRD after treatments show longer response and survival times [13,19]. Some evidence points to the existence of leukemic stem cells in CLL [5,6], although few studies are available, and these cells’ identity is not fully known. According to this model, despite the maturity characteristics of CLL B lymphocytes, the cells would originate from a more undifferentiated stage, which would generate monoclonal cells with defects in apoptosis mechanisms. The discovery and identification of stem cells in CLL would have great clinical and etiological relevance, as it would allow for predicting and monitoring the response to treatments more efficiently.

As a treatment strategy for CLL, increasing the levels of apoptosis in leukemic cells, especially in the most refractory ones, is a highly relevant objective, as it could be decisive in eliminating these residual reservoirs, minimizing MRD, and favoring more complete remissions [15]. Curcumin, a phenolic compound present in the rhizome of turmeric (*Curcuma longa*) and other plants from the Zingiberaceae family [20,21,22] is a molecule with pleiotropic effects, showing antimicrobial, anti-inflammatory and antioxidant activity, among others [20,21,23,24]. This compound has demonstrated an effect on different cancer progression pathways, high specificity, and low rates of resistance generation [25]. It also reduces the activation of different survival pathways and increases cancer cell apoptosis sensitivity. Moreover, it shows no toxicity in humans at effective doses, and a cytotoxic effect is achieved in leukemic cells at lower doses compared to non-leukemic cells [23,25]. Curcumin is a promising adjuvant therapy for CLL because it can increase apoptosis rates and potentiate the effects of other antitumor drugs, while having minimal toxicity to normal cells.

In this study, we provide novel evidence supporting the existence of stem-like cell populations in CLL, which could contribute to a deeper understanding of the disease as well as to development of more effective treatment options. We further evaluated the cytotoxic efficacy of curcumin against CLL cells and leukemic stem cells (LSCs) while examining its potential to potentiate the action of other drugs using both cell lines and primary cell cultures (the I83 cell line, originated from a CLL patient, and primary cell cultures derived from CLL patients). The I83 cell line was evaluated by isolating a rare diploid subpopulation from the tetraploid cell line through a limiting dilution technique. Immunophenotyping of CLL markers CD19, CD5, and CD38 as well as karyotyping and ploidy analysis were performed. We discovered that these cells possessed the remarkable ability to spontaneously regenerate the original tetraploid population, indicating their potential as stem-like cells. This characteristic enabled us to assess the effects of curcumin and various drugs on both CLL cells and leukemic stem cells (LSCs). To assess the effects of curcumin on primary CLL cell cultures, we utilized a functional analysis method previously developed by our group [11], enabling the identification of CLL cells with stem cell-like properties. This assay uses flow cytometry to detect alkaline phosphatase (ALP) activity, which is known to be highly expressed in pluripotent stem cells. ALP activity serves as an indicator of immaturity and pluripotency [26] and has been extensively studied to distinguish different stem cell subpopulations [27,28]. In these two models, we investigated the synergistic effect of curcumin in combination with fludarabine and ibrutinib, two drugs commonly used in the treatment of CLL, to target and eliminate potential leukemic stem cell populations. Fludarabine is a fluorinated nucleotide and purine analog that inhibits α-DNA polymerase, ribonucleotide reductase, and DNA primase, thus interrupting DNA synthesis. It is administered parenterally and usually used in combination with cyclophosphamide and rituximab, which is known as FCR [13,23]. Ibrutinib is a Bruton’s tyrosine kinase (BTK) inhibitor, which is essential in B lymphocyte maturation and is overexpressed in multiple B-cell cancers, conferring higher survival and proliferation rates. Ibrutinib is used as second line treatment to treat refractory CLL and is especially indicated in patients with 17p deletion [13,23]. Our findings demonstrate the efficacy of this combination therapy in eradicating these populations and may suggest its potential for improving treatment outcomes.

## 2. Results

### 2.1. Limiting Dilution of the I83 Tetraploid Cell Line Reveals a Diploid Subpopulation That Can Spontaneously Regenerate the Tetraploid Main Population

Limiting dilution of the I83 cell line followed by ploidy analysis using flow cytometry confirmed that a uniformly diploid subclone was obtained from the original tetraploid population (Figure 1A(I)) since all the cells analyzed were diploid. Immunophenotyping showed that both populations were positive for CD5, CD19, and CD38 (Appendix A).

This newly obtained subpopulation was expanded in cell culture. After 2 months, ploidy analysis showed the emergence of a secondary population (Figure 1A(II,III)), corresponding to 4n cells in G0-G1 and G2M phases, indicating the emergence of tetraploid cells originated from the diploid cell population. In the 6th month, tetraploid cells outnumbered the initial diploid population (Figure 1A(IV)). In the 9th month, ploidy analysis using flow cytometry detected only an apparently homogeneous tetraploid population, while the diploid population was undetectable (Figure 1A(V)). This tetraploidization process can be reproduced from a diploid subpopulation obtained by limiting dilution. However, cell culture expansion of a homogeneous tetraploid I83 population did not reveal the formation of diploid subpopulations. The tetraploid cell population exhibited significantly higher cell proliferation rates than the diploid population, contributing to its observed dominance and eventual replacement of the diploid clone over time.

Karyotyping revealed that these populations had a complex and heterogeneous composition, consisting of two main clones: a hypodiploid clone of 45 chromosomes (corresponding to the diploid population detected by flow cytometry) and a hypotetraploid clone of 88–90 chromosomes (corresponding to the tetraploid subpopulation detected by flow cytometry). The proportions of these clones were variable. The diploid subpopulation had 39 hypodiploid metaphases and 11 hypotetraploid metaphases (78% and 22%, respectively), while the tetraploid population had 27 hypotetraploid metaphases and 23 hypodiploid metaphases (54% and 46%, respectively). In both cases, structural deletions of 2p and 18p were detected (Figure 1B). The chromosomal formulas of the clones were 45,X,-Y,del(2)(p21p23),del(18)(p11.1) and 88–90,XX,-Y,-Y,del(2)(p21p23)x2,del(18)(p11.1) x2.

### 2.2. Diploid and Tetraploid I83 Presented Similar Susceptibility to Curcumin and Fludarabine

Analysis of cytotoxic effects of curcumin and fludarabine on diploid and tetraploid I83 revealed no significant differences in drug sensitivity, as was evaluated by cell membrane integrity. Comparing cells without treatment (control) to cells exposed to different concentrations of curcumin (5 µM and 10 µM) and fludarabine (1 µM, 5 µM, and 10 µM), we observed statistically significant differences in both diploid and tetraploid I83 cells (Appendix A).

### 2.3. ALP Activity Defines CLL Subpopulations with Higher Treatment Resistance

Peripheral blood samples from 25 patients with CLL were evaluated for the presence of populations with leukemic stem cell characteristics. Four and two different time points were studied in patients 4 and 6, respectively. Patient ages ranged from 53 to 88 years, with a mean age of 72 years. Eight female and 21 male patient samples were analyzed, which is consistent with the sex bias observed in the general CLL population. Clinically relevant data, including age, sex, risk factors, received treatments, and clinical history, were collected for all patients (Table 1).

Alkaline phosphatase activity, immunophenotyping, and membrane integrity were assessed in fresh peripheral whole blood and primary cell cultures derived from the whole blood. Three parameters were used to evaluate each sample: percentage of PI-negative cells, which is related to cell integrity; co-expression of CD19 and CD5 markers in the PI-negative fraction, which is used to identify the leukemic fraction; APLS fluorescence in the leukemic fraction, which corresponds with ALP activity. Since ALP activity is related to more primitive and immature cells [11,12,28], this fraction is useful for identifying potential leukemic stem cells (Figure 2).

Blood samples analyzed after extraction presented higher percentages of PI-negative cells than primary cell cultures (mean 92.01% and 45.04%, respectively), as could be expected due to lysis treatment and manipulation performed prior to cell culture establishment. Leukemic cell percentages of the PI-negative fraction slightly increased in untreated primary cultures (mean 52.97%, range 0.5–96.97%) compared with fresh blood samples (mean 40.42%, range 0.02–94.15%). Finally, an important enrichment in cells with high ALP activity was observed in the leukemic fraction. While in peripheral blood samples they only represented 27.38% of leukemic cells, in primary cultures they represented 85.57%.

### 2.4. Curcumin, Ibrutinib, and Fludarabine Show Efficacy to Reduce Population of CLL Cells with Highest ALP Activity

Primary cell cultures were established from the peripheral blood samples of 25 patients with chronic lymphocytic leukemia (CLL) under four different conditions: control (with DMSO): untreated cells; curcumin: cells treated with 5 µM curcumin alone; drug: cells treated with a single CLL drug (fludarabine, ibrutinib, rituximab, or venetoclax) alone; drug + curcumin: cells treated with a combination of a CLL drug and curcumin. The drug selection was based on the patient’s previous treatment. Primary cultures treated with fludarabine (*n* = 17) included patients who had not received previous treatment or who had only been treated with fludarabine or FCR (a first-line treatment regimen). Primary cultures treated with rituximab (*n* = 5) included patients who had been treated with rituximab. Primary cell cultures from two patients were prepared with both fludarabine and rituximab because they had received FCR as first-line treatment. Primary cultures treated with ibrutinib (*n* = 8) included patients who had been treated with a second-line treatment based on this drug. One sample was prepared with venetoclax, corresponding to one patient who had been treated with this drug (Appendix A). Results from the rituximab group were excluded due to an absence of response due to methodological limitations related to the in vitro use of this drug. Results from the venetoclax group were excluded due to an insufficient number of samples.

As shown in Appendix A, fludarabine group controls had high percentages of leukemic cells (mean 76%, range 31–97%) and relatively high levels of leukemic cells with high ALP activity (mean 82%, range 22–99%). Ibrutinib group controls had relatively low percentages of leukemic cells (mean 21%, range 1–32%) but uniformly very high levels of leukemic cells with high ALP activity (mean 96%, range 93–99%).

As described above, 17 primary cultures from 16 different patients were treated with 10 µM fludarabine. A total of 20 experiments were performed, as three samples (from patients P18, P21, and P22) were analyzed at two different incubation times. The combination of fludarabine and curcumin did not produce a significant synergistic effect in reducing the leukemic fraction compared to fludarabine alone (Figure 3A). However, the percentage of ALP-positive leukemic cells was significantly reduced with the combination of fludarabine and curcumin compared to fludarabine alone (Figure 3B), from 42.57% to 35.79%. The effect was additive, with a combination drug index (CDI) of 0.97. Curcumin alone had no significant effect on ALP activity compared to the control (Figure 3B).

Eight primary cultures from 5 different patients were treated with 10 µM ibrutinib. The combination of ibrutinib and curcumin did not significantly reduce the leukemic fraction compared to ibrutinib alone (Figure 3C). However, both curcumin and ibrutinib used alone significantly reduced the leukemic fraction compared to the negative control, from 21.33% to 14.22% (*p* = 0.023) and 6.72% (*p* = 0.016), respectively. The absence of a significant synergistic effect could be attributed to the limited sample size, as curcumin potentiated the effect of ibrutinib in all cases. It could also be attributed to the fact that ibrutinib already produced a strong effect when used alone. Curcumin clearly potentiated the effect of ibrutinib in all four follow-up samples derived from a single patient (Appendix A). For the effect on leukemic cells with higher ALP activity, the combination of ibrutinib and curcumin produced a potent and significant synergistic effect in eliminating them, compared to ibrutinib alone (percentage reduction from 62.93% to 19.23%, CDI = 0.37). Curcumin and ibrutinib also produced significant effects when used alone (Figure 3D).

## 3. Discussion

Chronic lymphocytic leukemia (CLL) is a disease with a high incidence in Western countries and a low remission rate [13,14,15,29]. Developed treatments have allowed many patients with CLL to live with the disease for a long time. However, these treatments can cause significant side effects and do not prevent the development of drug-resistant cells and disease progression in the long term, which require more complex and aggressive treatments. Leukemic stem cells are thought to be responsible for relapses and drug resistance in other leukemias, such as chronic myeloid leukemia and acute myeloid leukemia [3,4]. However, there is still no strong evidence to support their relevance in CLL. Identifying CLL leukemic stem cells as the main source of drug resistance and relapses would be highly clinically relevant.

We have identified an I83 diploid subculture that may be a model of CLL stem cells. The I83 cell line was derived from peripheral blood cells from a 75-year-old Caucasian man with CLL. These are Epstein–Barr virus-positive cells, with lytic infection and active virus production. Our results showed that both diploid and tetraploid populations were positive for the CLL markers CD19, CD38, and CD5, which is consistent with the available information on this cell line [30,31]. The I83 cell line has a hypotetraploid karyotype (83–88 chromosomes, <4n, XXYY) with a deletion in the long arm of chromosome 13 due to chromosomal rearrangements [30,31]. This information differs from our karyotyping results, which show that the tetraploid population has 88–90 chromosomes with structural deletions in the short arms of chromosomes 2 and 18. These differences could be due to the genetic instability of immortalized cell lines, combined with genetic drift and bottleneck effects due to limiting dilution. More research using I83 cells from different sources is needed to determine the variability of these cells. Nevertheless, the karyotyping results confirm the common origin of the diploid and tetraploid populations.

It is reasonable to hypothesize that the diploid population gives rise to tetraploid cells. Although the diploid population is rare, it would be maintained in a tetraploid culture and subsequently enriched by limiting dilution, allowing us to select a single diploid cell to generate a homogeneous subclone derived from the I83 cell culture. In stable conditions, tetraploid cells may have higher proliferation rates and may become most of the cell line, while diploid cells remain as a rare subpopulation that was initially undetectable by current techniques such as flow cytometry cell cycle and ploidy analysis.

One hypothesis derived from this observation is that the I83 cell line could maintain a nearly undetectable minority diploid subpopulation that could, nevertheless, be favorably selected in some conditions. The perpetuation of the diploid population at such low percentages suggests that it may have an essential function for the cell line, supporting the hypothesis that it is a potential stem cell model. This subpopulation would be in a strict minority and have the capacity to regenerate a wide progeny of tetraploid cells and eventually lead to disease. This hypothesis is consistent with the findings of Kikushige et al., who suggested that hematopoietic stem cells (HSCs) give rise to CLL [6]. Briefly, a xenograft of HSCs from a CLL patient that was grafted into mice revealed differences with healthy donors. CLL HSCs produced a higher percentage of B lineage cells and more frequently expressed CD5 than HSCs from healthy donors. Additionally, CLL HSCs generated characteristics more like monoclonal B lymphocytosis (MBL), a pre-malignant condition of CLL. Analysis of variable diversity joining (VDJ) recombination showed that the generated cells were different from those of the original patient and did not show the abnormal karyotypes associated with CLL. However, only CLL HSCs generated clonal B cells, and unlike control HSCs, they generated clones that were positive for CD5 and CD23, as well as a higher number of polyclonal B cells [6]. These results indicate that CLL is initiated in HSCs, but HSCs alone are not sufficient to cause the disease. HSCs would generate high numbers of B cells, some of which could acquire new mutations that drive the progression of MBL to CLL [6].

Tetraploidy is not a frequent condition in CLL, and its association with prognosis is not described in CLL ESMO guidelines [13]. In a 2017 study, it was found to be associated with aggressive disease characteristics: RAI stage 3/4, 17p deletion, complex karyotype, and ibrutinib discontinuation due to Richter transformation [32]. Moreover, diploid I83 cells may be a useful model for studying CLL disease progression. Tetraploidization was observed in two independent cell lines, suggesting that these cells tend to acquire this alteration. Tetraploidization may be a mechanism that confers higher resistance and proliferation rates, as it is observed in some CLL patients with disease progression. The reproducibility of the results suggests that cells may acquire specific mutations, which determine a hierarchical clonal evolution. Therefore, despite apparently being less aberrant, diploid cells could be the clonal origin of the disease and the residual disease that produces relapse and progression to more aberrant phenotypes after treatments generating apparent complete remission.

No significant differences were observed in the susceptibility of diploid and tetraploid I83 populations when treated with curcumin and fludarabine. This is consistent with the results of cell integrity analysis. More comprehensive comparative studies should be conducted using various drugs and combinations while considering additional parameters. Given the homogeneity of the cells in these populations, no significant differential responses could be discerned among cells; therefore, the effects on ALP activity or CD5 and CD19 expression were not assessed. More exhaustive analysis of this new subculture will be needed in upcoming studies to define its nature and response to multiple conditions.

We have also tested the ability of a functional analysis to detect CLL stem-like cells in peripheral blood samples. The results confirm that this method is effective in detecting highly resistant cells with potential leukemic stem cell function. This protocol is based on the elevated activity levels of alkaline phosphatase (ALP) in pluripotent stem cells. Our findings suggest that this functional analysis could be a valuable tool for detecting and monitoring CLL stem-like cells in patients. This protocol, based on the elevated activity levels of ALP in pluripotent stem cells [26,27,28], was successfully applied by our research group to identify refractory leukemic stem cells in acute myeloid leukemia [11,12]. These findings hold promise for future prospective studies, as they can be associated with clinical outcomes. The enrichment of cells with high ALP activity in primary cell cultures suggests that these cells are resistant to the stresses associated with sample manipulation, such as changes in pH, temperature, and nutrient availability. Our findings suggest that ALP activity may be a useful biomarker for identifying refractory CLL populations that are resistant to treatment. Further research is needed to validate this biomarker in a larger cohort of patients and to develop new therapies that target these refractory populations.

Reducing these populations is a priority for improving treatments. We evaluated the effect of curcumin, a widely studied phenolic compound with proapoptotic and chemopotentiating properties, in combination with fludarabine and ibrutinib. Curcumin has demonstrated activity through multiple pathways and mechanisms of action [23,24,33], including mechanisms related to drug metabolism and elimination, such as inhibition of CYP3A4 [34,35] and P-glycoprotein (P-gp) [35,36,37]; with inflammation, such as inhibition of COX-2 [38] and inhibition of NFκB activation [39,40,41]; or by inhibiting molecules and pathways related with cancer progression, reduced apoptosis, and cell survival, such as protein kinase C [42] and c-Jun/AP1 [43]. Despite its well-known antioxidant activity, it also seems to be capable of potentiating some drugs by producing a pro-oxidant effect on cancer cells [44]. Curcumin showed a strong synergistic effect when combined with fludarabine, and particularly with ibrutinib, leading to a decrease in the percentage of high-ALP CLL cells. This synergistic effect could have long-term implications, facilitating the elimination of refractory leukemic cells and enabling more prolonged and complete remissions.

In the ibrutinib group of samples, curcumin showed a significant selective effect in eliminating the leukemic fraction with high ALP activity when used alone. In contrast, it did not show a significant effect in the fludarabine group of samples unless combined with the drug. These differences may be attributed to the distinct nature of the studied samples. As previously described, samples from patients treated with ibrutinib exhibited lower percentages of leukemic cells, but they were enriched in cells with high ALP activity, indicating the selection of the most refractory cells resulting from the received treatments. This finding suggests that curcumin may be particularly effective in eliminating the most refractory leukemic cells, which would have clear beneficial effects.

Given that CLL cells have reduced apoptosis rates [15,16,17], curcumin emerges as a potential option to complement certain treatments and enhance their effectiveness. It can restore apoptosis levels and increase sensitivity to chemotherapy. Additionally, curcumin is selective for tumor cells and has low toxicity at effective doses [25,45,46,47,48]. Therefore, it can be administered to patients with comorbidities and pre-existing conditions, including chronic ailments.

A comprehensive analysis of patient parameters, including mutations, age, and sex, revealed no significant impact on their response to curcumin or conventional therapies. This lack of association is likely attributed to the relatively small sample size and the inherent heterogeneity among patients. To further elucidate potential associations between specific conditions and treatment response, larger-scale studies involving more individuals should be conducted. However, the present study has some limitations. First, cell cultures were conducted under normoxic conditions. This is the standard practice for in vitro studies, but the physiological conditions of cells in the body typically involve oxygen concentrations of approximately 2–10%. This difference can result in changes in gene expression, adhesion molecules, proliferation and survival rates, and drug resistance [49]. The most primitive and resistant cells, such as stem cells, are often found in the bone marrow, where lower oxygen concentrations prevail. This can lead to changes in gene expression and protein function, which may increase resistance to curcumin and drugs [50]. Additionally, proliferative CLL cells are primarily concentrated in the bone marrow and lymph nodes, while cells in circulating peripheral blood are typically in the G_0_/G_1_ phase [51]. Bone marrow samples are ideal for these studies because they contain the most primitive and resistant cells, such as stem cells. However, obtaining bone marrow samples can be challenging due to limitations in availability.

Our study made several important observations. First, we identified a potential cell model of CLL leukemic stem cells, providing valuable insights into these elusive cell populations. Second, we successfully adapted a functional analysis approach to effectively detect highly resistant populations with leukemic stem cell characteristics in CLL samples. Third, our study highlighted the effectiveness of curcumin as a drug adjuvant, specifically targeting and reducing these populations. This finding is significant because it sheds light on the limited understanding of CLL leukemic stem cells and their critical role in minimal residual disease, the development of relapse, and the chronicity of the disease.

## 4. Materials and Methods

### 4.1. Cell Lines

The tetraploid I83 cell line was kindly provided by Dr. Marta Crespo, Vall d’Hebron Institute of Oncology (VHIO), Vall d’Hebron Barcelona Hospital Campus. The diploid I83 population was obtained in our laboratory by limiting dilution of the original tetraploid I83 cell line in 96-well plates. One subculture, consisting exclusively of diploid cells, was expanded and used for the study. The clonality of the cultured cells was periodically assessed using flow cytometric analysis of the cell cycle distribution. Cells were cultured with RPMI supplemented with 10% FBS, 2% glutamine, 1% sodium-pyruvate, and 1% penicillin–streptomycin (Biowest, Nuaillé, France) at 37 °C and 5% CO_2_. Diploid I83 cells were cultured for nine months until spontaneous tetraploidization occurred.

### 4.2. Patient Sample Analysis and Primary Cultures

Peripheral blood samples (*n* = 29) were obtained from 25 patients with chronic lymphocytic leukemia (CLL) (17 males and 8 females; mean age of 72 years, range 53–88) using EDTA anticoagulated tubes from the Hematology service of ICO-Germans Trias i Pujol (Badalona, Spain). All patients gave their informed consent according to the Helsinki Declaration. All procedures were performed according to laboratory internal protocols, authorized by the Ethics Committee of Clinical Investigation of Hospital Universitari Germans Trias i Pujol (Badalona, Spain). Primary cultures were established directly from 300 µL of peripheral blood in 24-well plates. Erythrocytes were previously lysed using ammonium chloride lysis solution (1.5 M NH_4_Cl, 100 mM NaHCO_3_, 1 mM disodium EDTA, all purchased from Merck, Darmstadt, Germany, distilled H_2_O to 900 mL, pH to 7.4 with 1 N HCl or 1 N NaOH), diluting the sample with 1× lysis solution in a 1:10 ratio. After 10 min incubation at room temperature, samples were washed with Hanks’ Balanced Salts Solution (HBSS, Biowest) and resuspended in 300 µL of RPMI. Cells were seeded in a 6-well plate by adding 50 µL of cell suspension to 2 mL of RPMI supplemented with 10% FBS, 2% glutamine, 1% sodium-pyruvate, and 1% penicillin–streptomycin. Seeding leukocyte concentrations ranged from 40 to 4000 leukocytes/µL.

Primary cell culture treatments were determined by the patient’s treatment regimen. Conditions included controls with dimethyl sulfoxide (DMSO, Thermo Fisher, Waltham, MA, USA), curcumin alone, studied drug alone (fludarabine, ibrutinib, rituximab, or venetoclax), and studied drug combined with curcumin. Drugs, curcumin, or DMSO were added at 2 µL, resulting in final concentrations of 5 µM curcumin, 10 µM fludarabine and ibrutinib, 10 µg/mL rituximab, and 100 nM venetoclax. Plates were incubated at 37 °C and 5% CO_2_ for 2–6 days. Fludarabine was obtained from Aurovitas, Madrid, Spain; ibrutinib from Imbruvica, Janssen, Titusville, NJ, USA; rituximab from MabThera^®^, Roche, Basel, Switzerland; and venetoclax from Venclyxto, AbbVie, Chicago, IL, USA. A granulated curcumin formulation with 95% purity was obtained from Naturex (Avignon, France).

### 4.3. Immunophenotyping

Immunophenotyping characterization was performed using the following monoclonal antibodies: CD19-AF700, CD19-APC, CD19-PE, CD19-PE-Cy7 (clone LT19) (Sysmex, Kobe, Japan); CD5-APC, CD5-PacB (clone L17F12) (Sysmex); CD38-PE (clone HIT2) (Sysmex); CD5-PE (clone UCHT2) (Beckton Dickinson, Franklin Lakes, NJ, USA).

Briefly, cell count was performed to obtain the necessary volume containing 1 × 10^6^ cells. Cells were washed with HBSS and resuspended in 100 µL of HBSS with albumin and sodium azide (Sigma-Aldrich, St. Louis, MO, USA) to block non-specific binding between antibodies and markers. The required volume of monoclonal antibody was added following manufacturer indications, incubated at room temperature, and protected from light for 20 min. After incubation, 1000 µL of HBA was added and samples were acquired using a Invitrogen™ Attune™ NxT™ flow cytometer (Thermo Fisher).

### 4.4. Karyotyping

Conventional G-banding metaphases were obtained using standard procedures. Briefly, mitotic cells were arrested in metaphase with colcemid (Gibco, Waltham, MA, USA) and lysed in a hypotonic solution (0.075 M KCl). The metaphases were then fixed with Carnoy’s solution (3 parts methanol:1 part glacial acetic acid), and the G-banding pattern was obtained with Wright dye. Karyotypes for 20 metaphases were analyzed for each cell line and described following the International System for Human Cytogenetic Nomenclature [52].

### 4.5. Ploidy Analysis

To analyze ploidy using flow cytometry, cells were fixed and permeabilized. First, 1 mL of cell suspension (containing a minimum concentration of 5 × 10^4^ cells/mL) was centrifuged and the pellet was resuspended in 1 mL of 70% ethanol and incubated overnight at −20 °C. After this step, the ethanol was washed twice with HBSS and the pellets were resuspended in 1 mL of HBSS and incubated with 2 µL of DAPI 1 mg/mL (Sigma-Aldrich) for 30 min at room temperature in the dark. DAPI fluorescence was used to determine the percentage of cells in each cell cycle phase (G0/G1, S, and G2/M).

### 4.6. Comparative Cell Membrane Integrity Analysis on I83

To determine potential differences in drug sensitivity among diploid and tetraploid I83 cells, comparative analysis of the effects of curcumin and fludarabine on cell membrane integrity were performed. Cells were incubated at 37 °C and 5% CO_2_ with different concentrations of curcumin (5 µM and 10 µM) and fludarabine (1 µM, 10 µM and 20 µM). incubation time was 72 h. After incubation, 2 µL of PI 100 µg/mL were added, and the sample was incubated for 5 min at room temperature in the dark before flow cytometry analysis.

### 4.7. Alkaline Phosphatase Activity Analysis

To analyze the blood samples of CLL patients and primary cell cultures, four fluorescent labels were combined: Hoechst 33,342 (Invitrogen™ Waltham, MA, USA), a cell-permeant nuclear stain used to discriminate erythrocytes, platelets, and background from nucleated cells, avoiding lysis and washing steps [53]; propidium iodide (PI) (Invitrogen™) to discriminate necrotic cells; monoclonal antibodies conjugated to fluorochromes to detect CD19 + CD5+ cells; and Alkaline Phosphatase Live Stain (APLS) (Invitrogen™) to detect primitive and potential leukemic stem cells based on the alkaline phosphatase (ALP) activity. These markers were analyzed by adapting a protocol previously described by our group [11]. Briefly, whole blood containing 1 × 10^6^ nucleated cells or pellets obtained from each well in primary cell cultures were diluted with HBSS to a final volume of 100 µL. Then, 10 µL of FBS as a blocking agent and 10 µL of Hoechst 33,342 1 mg/mL were added, and the sample was incubated for 10 min at 37 °C in the dark. After incubation, 1 µL of APLS (stock concentration 500×) was added and each sample was incubated for 20 min at 37 °C in the dark. After incubation, CD5 and CD19 monoclonal antibodies were added and incubated for 20 min at room temperature in the dark. Finally, the sample was diluted with HBSS to a final volume of 3 mL. Then, 2 µL of PI 100 µg/mL were added, and the sample was incubated for 5 min at room temperature in the dark before flow cytometry analysis.

### 4.8. Data Analysis

Obtained results were analyzed using Attune™ NxT™ Software 2.6 version (Invitrogen™), Excel (Microsoft^®^ 2007, Redmond, WA, USA) and GraphPad Prism 9 version 9.0.1 (GraphPad Software, Inc., Boston, MA, USA). FCS Express 5 FlowResearch Edition version 5.01.0082 (De Novo Software™, Pasadena, CA, USA) was also used for cell ploidy analysis.

For primary cell cultures, the non-parametric Student’s *t*-test (Wilcoxon test) was used. Results with a *p*-value lower than 0.05 were considered significant. In all cases, values obtained for curcumin or drugs were compared with values obtained for the control to determine significant differences. Values obtained for drug combined with curcumin were compared with values obtained for drug without curcumin. To calculate the synergistic effect of curcumin, the coefficient of drug interaction (CDI) was calculated using the following formula:$$CDI=\frac{\%\;of\;cells\;(Drug\;+\;Curcumin)}{\%\;of\;cells\;\left(Drug\right)\;\times \;\%\;of\;cells\;(Curcumin)}\;\times \;100$$

CDI < 1 indicates synergism.CDI = 1 indicates additive effect.CDI > 1 indicates antagonism.

## Figures and Tables

**Figure 1 ijms-25-01994-f001:**
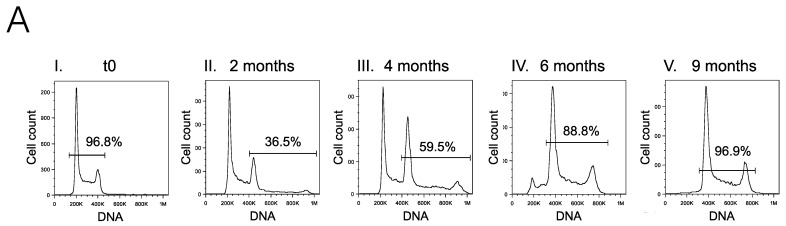
I83 subclone ploidy analysis after limiting dilution from an original tetraploid I83 cell line. (**A**) Flow cytometric analysis of ploidy and cell cycle distribution displaying I83 tetraploidization. The obtained I83 subpopulation derived from one single cell clone of a tetraploid I83 cell line was initially uniformly diploid. A small secondary G2M peak was detected on the 2nd month and was indicative of the emergence of a tetraploid cell clone derived from the diploid cell line. Over an approximately nine-month period, the tetraploid population exhibited a progressive increase, leading to the eventual replacement of the initial diploid population due to its higher proliferation rate. The diploid population eventually became undetectable. (**B**) Karyotype of I83 cells. Karyotyping of hypodiploid subpopulation showed 45 chromosomes, presenting Y deletion. Arrows indicate 2p and 18p deletions. Hypotetraploid subclone (88–90 chromosomes) presented duplication of chromosomes, maintaining the same structural mutations.

**Figure 2 ijms-25-01994-f002:**
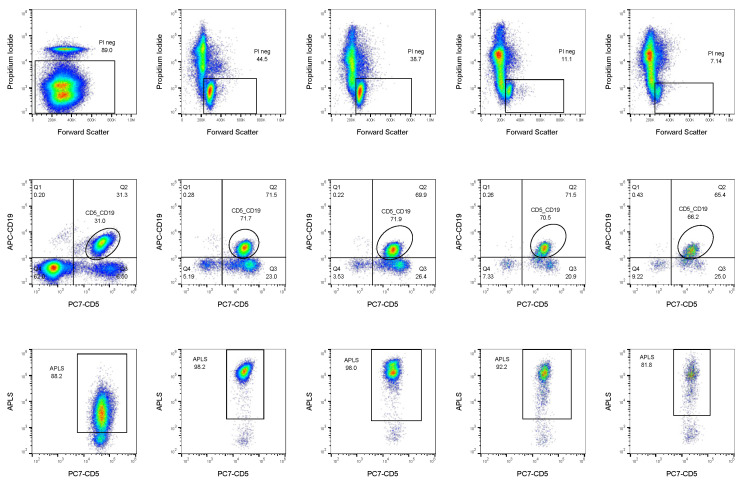
Strategy of analysis of samples and primary cell cultures. Plots exemplifying data obtained at the flow cytometer. For every patient, 5 different analyses, corresponding to columns, were performed: time zero, directly analyzing peripheral blood; and control and treatments with curcumin, drug, and drug plus curcumin, analyzing primary cultures after incubation with corresponding treatments. For every analysis, three parameters corresponding to rows were obtained: cell membrane integrity (upper row), CD19 and CD5 expression (middle row), and ALP activity (lower row).

**Figure 3 ijms-25-01994-f003:**
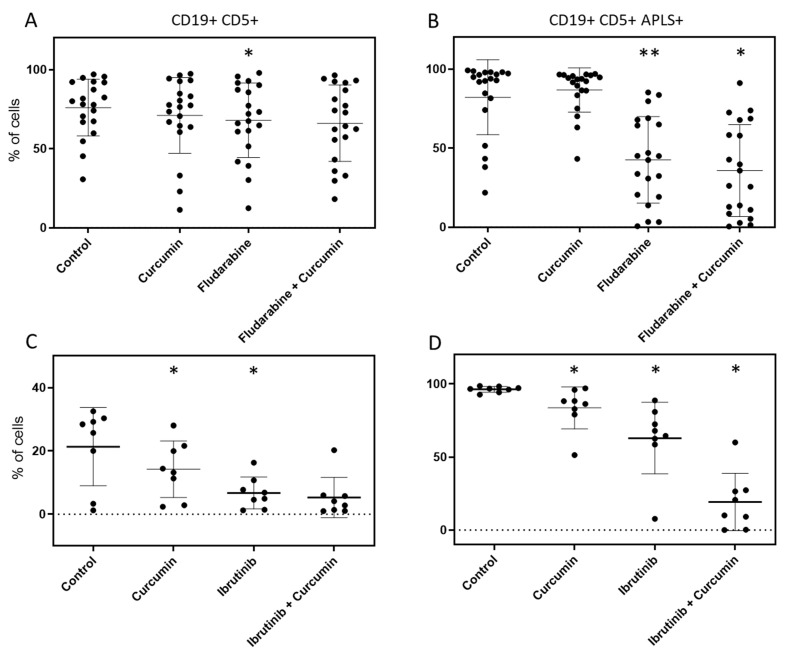
Curcumin synergistic effect combined with fludarabine and ibrutinib on CLL primary cell cultures. Primary cultures were directly established from CLL patients’ peripheral blood and incubated at 37 °C in 5% CO_2_ with different treatments. Curcumin, fludarabine, and ibrutinib concentrations used were 5 µM, 10 µM, and 10 µM, respectively. Incubation time ranged from 2 to 6 days. (**A**) Curcumin showed no significant effect on the leukemic fraction of samples of the fludarabine group, nor alone, nor combined with the drug. Fludarabine showed a significant effect in reducing the leukemic fraction. (**B**) Fludarabine showed a strong significant effect in reducing leukemic cells with high ALP activity, and combination with curcumin showed a significant synergistic effect in increasing this reduction. Curcumin did not show a significant effect when used alone. (**C**) Curcumin showed no significant synergistic effect combined with ibrutinib on the leukemic fraction, which was strongly reduced by the drug. However, curcumin significantly reduced the leukemic fraction when used alone. (**D**) Curcumin and ibrutinib showed q significant effect in reducing leukemic cells with high ALP activity in the ibrutinib group. Curcumin produced a strong and significant synergistic effect when combined with ibrutinib. Analysis was performed using flow cytometry. Among cells with intact cell membranes (PI-negative), cells positive for both CD19 and CD5 were selectively identified (leukemic cells). From these, cells with high ALP activity (stem-like) were selected (**B**,**D**). Substantial interpatient variability is associated with clinical differences among patients. Results were statistically analyzed using non-parametric Student’s *t*-test (Wilcoxon test). Values obtained for curcumin or drugs are compared with values obtained for the control. Values obtained for drug combined with curcumin are compared with values obtained for drug without curcumin. ** p <* 0.05; *** p <* 0.001.

**Table 1 ijms-25-01994-t001:** Clinical data of studied CLL patients.

	Timepoint	Leukocyte Count (×10^6^/L)	Age	Sex	% CD19 + CD5+	% APLS + (over CD19+ CD5+)	Status	Treatment	Prognostic Factors
P1		5.3	72	M	0.776	11.673	Partial response, MRD 0.03%	Ibrutinib + Ofatumumab	86% of nuclei with del(13)(q14) (DLEU) and del(11)(q22) (ATM)
P2		1.9	71	M	0.02	25	Complete remission	Rituximab-Bendamustine	Del(13q), mutated IGHV
P3		12.5	63	M	28.042	0.804	Leukocytosis since 2012, CLL features present	R-CHOP (6 cycles)	Cytogenetics:+13; t(14;19). Transforming CLL, no Richter criteria
P4	1	7.6	65	M	9.917	6.319	Partial response, MRD 35%	Ibrutinib, alone and combined with Ofatumumab	Del(13)(q14), del(11q) (ATM), ZAP70+. BinetB, Rai1
	2	7.8	65	M	8.647	2.453	Partial response, MRD 33%		
	3	7.4	65	M	14.616	6.495	Partial response, MRD 29%		
	4	8.6	65	M	1.597	0.158	Partial response, MRD 28%		
P5		13	70	F	55.234	85.899	MRD 55%.		Del(11)(q21q25) (ATM), del(13) (q14q21) (DLEU)
P6	1	11.8	79	M	46.745	0.343	Lymphocytosis, 86%; lymphocytes with CLL pheotype, narrowly suprassing 5000/µL; follow-up needed to confirm CLL	None	
	2	16.7	79	M	17.287	0.052	Binet-RaiA1. MRD 87%, CLL confirmation		
P7		128	83	F	82.265	93.923		None	
P8		13	80	F	31.113	87.753	Diagnostic	None	
P9		26.4	80	F	73.914	70.039	Progressive lymphocytosis; 84% lymphocytes with CLL phenotype. Binet III, Rai3 (lymphocytosis + anemia), relapse-progression	Tx1: Obinutuzumab + Chlorambucil. Tx2: Ibrutinib (intolerant). Tx3: R-idealisib (intolerant). Pending of Venetoclax	t(12;13)(q14;15), del(13)
P10		5.7	74	M	1.3	5.221	Partial response, 68% of B lymphocytes with CLL phenotype. RaiIV Binet C. Almost complete remission	Ibrutinib	Unmutated IGHV
P11		7.1	74	F	26.492	5.236	Incipient relapse, 48% of CLL lymphocytes in peripheral blood. Brother with CLL	Tx1: R-CHOPx6 and TITx6, radiotherapy. AutoTPH	Del(13q)
P12		10	66	M	44.491	1.911	69% CD19/CD5	None	
P13		13.6	67	M	27.716	34.293	CD19/CD5 51%. Recent diagnostic	None, control follow-up	
P14		7.9	56	M	21.358	85.838	Relapse, stage IV, 25% MRD in peripheral blood	Tx1: R-FC (2010), Tx2: R-idealisib (2016), Tx3: ibrutinib (2016), starting venetoclax	ZAP70+, del(13)(q14) (DLEU). Altered karyotype, del TP53
P15		154	77	M	92.646	88.506	Progression. CD19/CD5 lymphocytes 91%		FISH: del(13)(q14)(DLEU), del(17)(p13)(TP53) in 23% nuclei
P16		13,800 *	81	M	54.798	96.777	Relapse. MRD: 90% in peripheral blood		Del(11)(q22) (ATM gene)
P17		10.9	71	M	53.99	2.516	MRD: 85%. Incipient relapse	R-FC	Del(13)(q14)(D13S319)
P18		89.5	79	M	94.154	0.247	Progression. CD19/CD5 lymphocytes 95%		Del(13)(q14)(DLEU)
P19		13.2	71	F	69.002	82.096	Diagnostic, 75% CD19/CD5 BinetA, Rai1	None	
P20		24,600 *	53	F	64.385	0.028	Progression. MRD: 75% in peripheral blood		FISH: 92% of biallelic del(13)(q14) (DLEU)
P21		114	88	M	86.879	0.192			Trisomy 12.
P22		16.3	79	M	51.03	0.034	Diagnostic, 75% CD19/CD5		
P23		18.2	73	M	55.723	0.069	Diagnostic, 74% CD19/CD5 A1 Stage	None	Trisomy 12, del(13)(q14) (DLEU)
P24		10.5	81	F	3.887	0	Diagnosed on 2011. Progression in 2017. Lymphocytes with CLL phenotype: 40%	Tx1: Obinutuzumab + Chlorambucil. Tx2: Ibrutinib (2018)	ZAP70+, trisomy 12. 4.3 beta 2 microglobulin (September 2019)
P25		78	62	M	54.284	0.016	Diagnostic. MRD: 85%in peripheral blood		

In these two patients the count was obtained as lymphocytes/µL (*).

## Data Availability

The data that support the findings of this study are available from the corresponding author, J.P., upon reasonable request.

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
