# Peer review of "Persistence of Chronic Lymphocytic Leukemia Stem-like Populations under Simultaneous In Vitro Treatment with Curcumin, Fludarabine, and Ibrutinib: Implications for Therapy Resistance"

_ijms, 2024, doi:10.3390/ijms25041994_

Round 1
Reviewer 1 Report
Comments and Suggestions for Authors
In this manuscript Bistué-Rovira À et al. suggest that an I83 diploid subculture could be a model for CLL stem cells. They suggest using diploid I83 cells as a model for studying CLL progression. However, this point has not been linked to the next part of the work. The functional analysis to detect CLL stem-like cells in peripheral blood samples seems like a completely separate task.
I have the following comments and questions:
1. Section 3.2.: The authors should relate the obtained results to CLL patients' clinical data (e.g. prognostic factors, received treatment and clinical history).
2. Do you have longitudinal results of the same patient (e.g. P4 and P6)? Before and after treatment?
3. Have the authors performed a similar functional analysis with diploid I83 cells?
4. Labeling all panels in Figure 1 is necessary. I am unable to locate panels A and B. Only markings I, II, III, and IV are available.
Reviewer 2 Report
Comments and Suggestions for Authors
In this work Bistué-Rovira and colleagues explore the spontaneous generation of tetraploid subpopulation from diploid I883 CLL cells (first part). Then, they employ a different model (CLL primary samples; second part) to test the ALP activity and related it with a stem-like phenotype to finally evaluate the effect of conventional chemotherapeutics in combination with curcumin. The manuscript is well organized, and the figures are acceptable, however many aspects must be discussed in order to make this manuscript suitable for publication.
One of my main concerns is that I observe no direct relation between the first and second part of the manuscript. additionally the synergistically effect of curcumin in leukemic cells have been previously observed for other leukemic cells types in which the mechanism is fully explored, however despite the presented data, the authors barely mention or hypothesized how does curcumin synergize with the evaluated drugs. As a main comment
Is any report of the ploidy found in relapsed CLL patients?
L135 about the exclusively diploid cell culture: How does the authors confirm the purity of the diploid culture?
Does the authors monitor the cell growth of diploid vs tetraploid CLL cultures? Is there any difference in growth patterns or metabolism? Is there any possibility that “diploid” Figure 1 I, culture have a minimal tetraploid cell population? That has greater proliferative capacity compare to diploid cells, and as a competitive process they appear with time?
2.1: How was pH controlled in cell culture?
2.1/2.2: why does the authors decided to culture the cells in glucose-free media? The adaptations in cell metabolism under such conditions must be carefully attended. E.g., glucose deprivation is a commonly used to induce autophagy, a condition that’s further sensitizes cells to multiple drugs. This must be discussed.
2.2: Reagents item must be stated
Figures: some figures employed roman numbers to indicate figure content I, II, III whereas other employ letters. Please homogenize.
Figure 1.I T0 population has 2 peaks, whereas III, IV, exhibit 3 peaks, how does the authors discuss such data?
Figure 1. Are these representative data? The authors may want to quantify and compare statistically the proportion of spontaneous tetraploid population across time.
Figure 2. How does the authors justify adequate culture conditions if almost half of the initial population is lost during the first 72 h of culture in absence of any treatment?
In figure legends: incubation time and drugs concentrations must be stated
Figure 3: experimental approach must be defined as well as drug concentration employed and incubation time. As well as the statistical approach employed. And cell model.
Figure 3. if they represent primary CLL cultures, where they tetraploid?
Figure 3. “(d) Curcumin and ibrutinib showed significant effect reducing leukemic cells with high ALP activity.” This statement is only partially supported by the results and not conclusive, as curcumin did not exert any effect in (B). same observation applies for L446
Figure 3. Why some control cells have 20% whereas other 100%, 75% etc. the authors must explain the great variability, are they living cells? Which method was employed to determine such data.
L234 “Immunophenotyping showed that both populations were positive for CD5, CD19, and CD38” This results must be presented.
L21 et al. and other Latinisms must be italicized.
L71 CLL was previously defined (L55)
Sometimes Fig. X is used to reference a figure whereas other Figure X, please homogenize.
Comments on the Quality of English Languageenglish is fine
Round 2
Reviewer 1 Report
Comments and Suggestions for Authors
Authors have sufficiently addressed most of my concerns.
Reviewer 2 Report
Comments and Suggestions for Authors
most of my comments were considered. I have not further suggestions